# A Novel Method of Generating Geospatial Intelligence from Social Media Posts of Political Leaders

**Fahim Sufi** [1,*] and **Musleh Alsulami** [2]

1    Independent Researcher, Melbourne, VIC 3000, Australia
2    Information Systems Department, Umm Al-Qura University (UQU), Makkah 24382, Saudi Arabia; mhsulami@uqu.edu.sa
*    Correspondence: research@fahimsufi.com

**Abstract:** Social media platforms such as Twitter have been used by political leaders, heads of states, political parties, and their supporters to strategically influence public opinions. Leaders can post about a location, a state, a country, or even a region in their social media accounts, and the posts can immediately be viewed and reacted to by millions of their followers. The effect of social media posts by political leaders could be automatically measured by extracting, analyzing, and producing real-time geospatial intelligence for social scientists and researchers. This paper proposed a novel approach in automatically processing real-time social media messages of political leaders with artificial intelligence (AI)-based language detection, translation, sentiment analysis, and named entity recognition (NER). This method automatically generates geospatial and location intelligence on both ESRI ArcGIS Maps and Microsoft Bing Maps. The proposed system was deployed from 1 January 2020 to 6 February 2022 to analyze 1.5 million tweets. During this 25-month period, 95K locations were successfully identified and mapped using data of 271,885 Twitter handles. With an overall 90% *precision*, *recall*, and $F_1$ *score*, along with 97% *accuracy*, the proposed system reports the most accurate system to produce geospatial intelligence directly from live Twitter feeds of political leaders with AI.

**Keywords:** sentiment analysis on political tweets; named entity recognition on political tweets; geospatial intelligence; analyzing tweets of political leaders; big data processing of social media

## 1. Introduction

As of July 2021, 5.2 billion people have access to the internet, representing 65% of the global population [1]. Many of these people have access to social media and are constantly posting updates on their social media accounts. Hence, in every minute Facebook users share 240,000 photos, Facebook live receives 44,000,000 views, Instagram users share 65,000 photos, YouTube users stream 694,000 h of videos, Snapchat users send 2,000,000 Snapchats, and Twitter users post 575,000 tweets, generating a massive volume of big data [1]. A study in 2020 found that there are 1089 active Twitter accounts of heads of states, ministers, diplomats, and political leaders out of which 632 have been verified by Twitter [2]. Derogatory remarks in social media by political leaders are reacted to and retweeted since the public is drawn to negative events and news reports [3]. For example, A tweet by Donald Trump at 9:43 p.m. on 20 July 2020 states, "We are united in our effort to defeat the invisible China virus, and many people say that it is patriotic to wear a face mask when you can't socially distance. There is nobody more patriotic than me, your favorite president" [2]. This tweet almost instantly reached 20 million or 23% of the total followers (i.e., 88 million) of Donald Trump [2]. From then, the followers retweeted, reacted, and shared this view, creating a global impact on the country mentioned within the tweet. Measuring the impact of social media posts on a particular nation, country, or state is significant for any nation since diplomatic ties and geopolitical situations are severely affected by derogatory and negative posts by political leaders. Even though there has been

multiple politically driven research conducted with Twitter feeds [4–7], there has not been any significant study reported on measuring the impact of negative comments by a political leader about any geographic location.

It is methodologically possible to perform behavioral analysis of Twitter users by harnessing big data extracted from their social media accounts [8] and then using various artificial intelligence (AI)-based techniques reported in our earlier studies [9–18]. Existing studies in the political science domain either used geotagged tweets or completely skipped usage of location extraction algorithms, as observed in [4–7,19,20]. For example, in [19], geotagged tweets, in which location information already exists in the tweets, were only used for analysis. On the other hand, in [7], location information was never utilized, even though the study in [7] concerned the political domain. The existing corpus of political sciences never utilized sophisticated location extraction mechanisms such as named entity recognition (NER) that can generate location information when the geotagged location field is absent within the Twitter feed.

In this paper, a methodology is proposed for extracting social media posts from verified Twitter accounts of political leaders for subsequent analysis with sentiment detection, NER, and geospatial intelligence algorithms. This methodology was used on Twitter data captured from 1 January 2020 till 6 February 2022, on more than 1.5 million tweets of regional political leaders, heads of states, diplomats, foreign correspondents, and supporters of political parties, as well as political news agencies. More than 95,000 unique locations were extracted from these tweets using NER, and corresponding sentiment analysis was superimposed on geographic maps (both ESRI ArcGIS Maps and Microsoft Bing Maps). This process created hundreds of maps with thousands of locations demonstrating positive, negative, or neutral views of leaders on a particular area. Moreover, heat maps were generated, and hot-spot analyses were performed using Getis-Ord Gi*. Lastly, we deployed the final application (i.e., the proposed new software tool) on multiple platforms and on a range of devices covering Windows, iOS, and Android Apps. By using the outcome of this research, a political scientist would obtain a greater level of geographic intelligence than that obtained from the outcome of previous studies in political science [4–7,19,20].

To the best of our knowledge, extracting location-centric views of political leaders using artificial intelligence (AI)-based NER, sentiment analysis, and natural language processing (NLP) was never reported in the existing literature. Moreover, the following are some selected outcomes reported in this study:

- This study reported the longest interval of monitoring political tweets from 1 January 2020 till 6 February 2022;
- This study represented the largest collection of tweets, belonging to 271,885 political leaders, diplomats, government officials, and supporters of political parties, compared with existing studies in [4–7,17,21,22];
- This study represented the largest collection of multilingual tweets, covering 63 distinct languages. Of these different languages, 578,706 tweets were in Arabic, 320,221 tweets in English, 71,983 tweets were in French, 65,430 tweets were in Farsi/Persian, 29,103 tweets were in Spanish, and 22,219 tweets were in German, among many others;
- This study recorded the largest collection of locations, with over 95K locations automatically extracted using NER and mapped on both ESRI ArcGIS Maps and Microsoft Bing Maps;
- Most importantly, the proposed system demonstrated 97% overall *accuracy*, making it the most accurate geospatial system on processing tweets of political leaders [4–6,17,21,22].

## 2. Background

Over the last several years, social media platforms have been efficiently and effectively used by the governments and leaders of 189 countries, representing more than 98% of member states of the United Nations except for Laos, North Korea, and Sao Tome, as well as Principe and Turkmenistan [2]. Twitter, as the most prominent social media platform for

politicians, has been used for election campaigns [21]. Since political parties, candidates, and supporters use Twitter to disseminate political agenda, objectives, and visions effectively and efficiently, it is possible for data scientists to exploit the publicly available Twitter feeds to predict election results using sophisticated techniques, as demonstrated in [5]. In [5], sentiment analysis, classification algorithms such as support vector machine (SVM) were used, along with social network analysis (i.e., who is actively following and reacting to the Twitter handles of main electoral candidates) for predicting US elections. Moreover, locations of Twitter users supporting sensitive political agendas and issues (i.e., who is supporting and who is opposing a particular political agenda) have been demonstrated in [4]. In [4], sensitive political agenda such as the Citizenship Amendment Act (CAA) of India was monitored on the Twitter platform with sentiment polarity analysis to determine the locations of supporting or opposing views. In more recent research studies, we have used sentiment analysis and NER with innovative algorithms to identify activities and locations of sociopolitical anti-vax and pro-vax groups [17]. Previous studies such as Refs. [4,5,17,21] collected location-based political messages from Twitter using the location field of tweets [4]. The disadvantage of utilizing the location field of Twitter is that users are inclined to move around, and tweets can preserve an invalid address of the Twitter user. Furthermore, if location services are switched off by a Twitter user, analysis using the location field of the Twitter feed might result in an inaccurate outcome. Nevertheless, a location-oriented Twitter feed is important since research in [23] has demonstrated that combining location features with the sentiment analysis process offers enhancement to tweet sentiment classification.

Other than researchers and data scientists using Twitter feeds of political messages, many foreign government entities such as Russia often use influence bots to create socioeconomic conflicts for attaining strategic benefits [6,22]. Propaganda supported by refined technology (e.g., influence bots) is often used to craft unrest in social and political areas. As an example, recent inquiries by the US State Department and the UK Foreign Office have briefed how Russian intelligence entities discredit Western COVID-19 vaccines by conducting information operations that implied vaccination turning people into monkeys [22]. Realizing this highly complex challenge of mitigating social and political conflicts created by social media campaigns, cyber propaganda, and information operation, in 2015, the US Defense Advanced Research Projects Agency (DARPA) requested scientists and researchers to pinpoint "influence bots" on Twitter in a stream of tweets, concentrating on finding the actor behind these information operations [6]. Within that bot challenge, researchers exposed direct links to Russian troll accounts connected to companies supported by the Russian governmental agencies that excel in creating cyber influence [6]. Moreover, it was exposed that "93% of tweets about vaccines are generated by accounts whose provenance can be verified as neither bots nor human users yet who exhibit malicious behaviors" [6]. Foreign governments such as Russia often use influence bot to generate socioeconomic tensions, as well as political conflicts, for serving their own strategic interests [6,22]. It should be mentioned that none of the existing studies briefed in this section [4–6,17,21,22] have extracted location-centric views of political leaders on a range of geospatial threat maps.

From an ethical perspective, several research studies have been conduced on how social media users feel about their data being used by researchers or political scientists [24,25]. According to these studies, public-facing open social platforms such as Twitter have negligible ethical concerns since users of these platforms are already aware that all their posts can be openly viewed by the wider public, organizations, or entities. However, there could be a higher degree of ethical and privacy concerns over using data from Facebook, since with multilevel privacy features, Facebook users believe that their data could only be accessed by their closed social network. On the other hand, a study in [25] concludes that even though Twitter users do not have any issues with their data being used by researchers or scientists, they certainly do not want government and intelligence agencies to spy on their social media contents. Hence, in this research, we used publicly available Twitter feeds of political leaders to generate AI-based geospatial intelligence for the benefit of political scientists and social scientists.

## 3. Materials and Methods

Twitter-based diplomacy study, referred to as Twiplomacy 2020, revealed thousands of Twitter handles of political leaders [2]. Using several of these handles, our system used application programming interfaces to retrieve all the political tweets from 1 January 2020 till 6 February 2022. Our system was implemented with language detection, as well as translation Application Programming Interfaces (APIs) of Microsoft Cognitive Service, which supports more than 110 different languages [26,27]. Using these automated APIs, our system translated all non-English tweets (i.e., almost 1.43 million) into English. Subsequently, sentiment analysis was performed on all 1.5 million tweets. The sentiment analysis process returns the sentiment confidence of a particular tweet being positive, negative, neutral, or mixed sentiment. Finally, NER was performed on all tweets. Our system automatically extracted 24 different types of entities from which it automatically clustered 5 types of location-oriented entities (i.e., city, continent, country region, language, and state) with aggregated sentiments. The entire process is depicted in Algorithm 1 as pseudocode.

The system proposed in this paper can work with any other social media platforms, such as Facebook, Instagram, and LinkedIn, via supported APIs with Microsoft Power Platform, as depicted in Figure 1. Moreover, this system can also extract dynamic content from websites by using web scraping technology supported by Microsoft Power Platform (i.e., by using M language) [11,12]. As seen from Figure 1, MS Power Automate [28] and MS Power Query were used for acquiring tweets, data cleansing and transformation, language detection and language translation [26,27], sentiment analysis, and finally, NER. These preprocessed messages were then stored and managed in Azure Cloud hosted Microsoft SQL Server. SQL queries were used for data exploration, further analysis, and serving dashboards with filtered queries. Finally, Microsoft Power BI was used for data visualization, analysis, generating insights with AI and NLP [9–12,17]. Data Analysis Expression (DAX) programming language used in Microsoft Power BI was used for fetching filtered rows from MS Azure SQL Server, as shown in Code 1. Within MS Power BI, DAX can also be used for directly calling NER API from cloud-based Microsoft Azure Cognitive Services, as shown in Code 2. It should be mentioned that, in this research, both sentiment analysis and NER were performed with Microsoft Cognitive Services Text Analytics API [29].

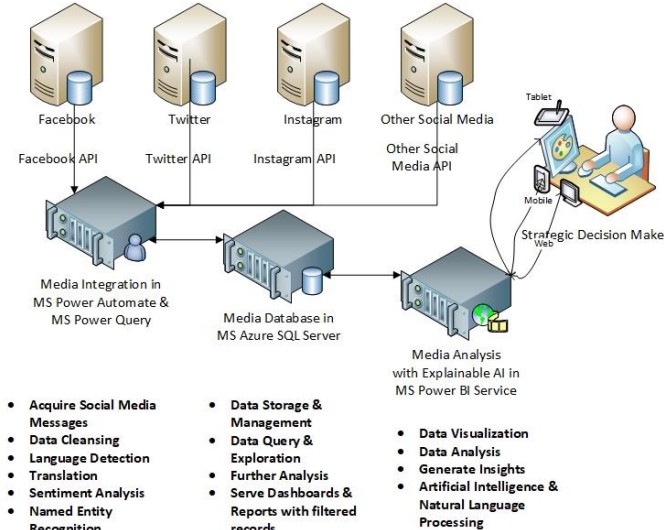

**Figure 1.** High-level architectural diagram of social media acquisition and geospatial intelligence processing from posts of political leaders.

As mentioned before, while existing research in political science on Twitter data used geotagged location field of Twitter data, this study generated new locations information, even if geotagged location information is missing. Using NER in a unique mechanism, demonstrated with Algorithm 1, Code 1, and Code 2, we generated location information and created heat

maps demonstrating political threats and opportunities. Since our algorithm generates new location information, we needed to assess the *accuracy* of newly generated location information.

To assess the end-to-end performance of our algorithm, we used several evaluation metrics, including *accuracy* (i.e., the fraction of the total samples that were correctly classified by a classifier), *precision* (i.e., the fraction of predictions as positive categories that were actually positive by the classifier), *recall* (i.e., the fraction of all positive samples that were correctly predicted as positive by the classifier), and the $F_1$ *score* (i.e., *precision* and *recall* combined into a single measure) as represented by Equations (1)–(4).

$$Accuracy = \frac{TP+TN}{TP+TN+FP+FN} \tag{1}$$

$$Precision = \frac{TP}{TP+FP} \tag{2}$$

$$Recall = \frac{TP}{TP+FN} \tag{3}$$

$$F_1\ Score = \frac{2 \times Precision \times Recall}{Precision+Recall} = \frac{2 \times TP}{2 \times TP+FP+FN} \tag{4}$$

---

**Algorithm 1:** Create location-centric sentiments for selected keywords from social media feeds of political leaders

---

| | |
|---|---|
| **Input:** | *Incoming Social Media Message,* $T = \{t_1, t_2, t_3, \dots\}$, *List of Handles as Keywords,* $K = \{w_1, w_2, w_3, \dots\}$ |
| **Output:** | Sentiment on the topic of the keywords on each country mentioned within the input Social Media Messages |

1.    **For Each** *tₙ ∈ T, n = 1 to |T|*
2.        **For Each** *Wₘ ∈ K, m = 1 to |K|*
3.            **If** *Wₘ ∈ tₙ Then*
4.                *sᵢ = tᵢ//Add tₙ to S (i.e.,* $t_n \in S$)
5.            **End If**
6.        **Loop**
7.    **Loop**
8.    **For Each** *sᵢ ∈ S, i = 1 to I*
9.        **If** *Detect_Language(sᵢ)= 'en'*
10.           *eᵢ = sᵢ//Add* $s_i$ *to E (i.e.,* $s_i \in E$)
11.       **Else**
12.           *eᵢ= Translate_English(sᵢ)//Add Translate_English(sᵢ) to E (i.e.,* $Translate\_English(s_i) \in E$)
13.       **End if**
14.   **Loop**
15.   **For Each** *eᵢ ∈ E, i = 1 to I*
16.       **If** *Detect_Entity(eᵢ)= 'Location'*
17.           *Add eᵢ to F (i.e.,* $e_i \in F$)
18.       **End If**
19.   **Loop**
20.   **For Each** *fⱼ ∈ F, j = 1 to J*
21.       $b_j = Get\_Locations(f_j)$
22.       $h_j = Analyse\_Sentiment(f_j)$
23.       **For Each** *location, cₙ in bⱼ*
24.           $C = \begin{cases} x: x = \emptyset, if\ c_n \exists C \\ x: x = c_n, if\ c_n \not\exists C \end{cases}$   *(i.e., Add Cₙ to locations master list C, only if C does not already include Cₙ)*
25.       **Loop**
26.   **Loop**
27.   **For Each** *cₙ ∈ C, n = 1 to N*
28.       **For Each** *bⱼ ∈ B, j = 1 to J*
29.           *Collate a list of M number of Sentiments hⱼ where cₙ ∈ bⱼ to compute:* $y_n = \frac{\sum_{m=1}^{M} h_m}{M}$
30.       **Loop**
31.   **Loop**

---

**Code 1:** DAX code for retrieving filtered tweets from SQL server

---

```
1.  let
2.      Source = Sql.Database("drsufiserver.database.windows.net", "SUFITWEETDB", [Query =
        "SELECT * FROM [dbo]. [Tweets] WHERE TweetSourceType like '%Regional Political
        Leader%'", CreateNavigationProperties = false]),
3.      #"Added Conditional Column" = Table.AddColumn(Source, "MasterTweet", each if
        [TweetLanguage] = "en" then [TweetText] else if [TweetLanguage] <> "en" then
        [TranslatedText] else null),
4.      #"Filtered Rows" = Table.SelectRows(#"Added Conditional Column", each ([TwitterHandle]
        = "BA_Yildirim") and ([Sentiment] = "negative")),
5.      #"Invoked Custom Function" = Table.AddColumn(#"Filtered Rows", "EntityDetection", each
        EntityDetection ([MasterTweet])),
6.      #"Expanded EntityDetection" = Table.ExpandTableColumn(#"Invoked Custom Function",
        "EntityDetection", {"name", "wikipediaScore", "text", "offset", "length", "entityTypeScore",
        "wikipediaLanguage", "wikipediaId", "wikipediaUrl", "bingId", "type"}, {"name",
        "wikipediaScore", "text", "offset", "length", "entityTypeScore", "wikipediaLanguage",
        "wikipediaId", "wikipediaUrl", "bingId", "type"})
7.  in
8.      #"Expanded EntityDetection"
```

---

**Code 2:** DAX code for calling NERAPI of Microsoft Cognitive Services Text Analytics

---

```
1.  (text) => let
2.  apikey       = "##API-Keys-Goes-Here##",
3.      endpoint     = "https://uaenorth.api.cognitive.microsoft.com/text/analytics/v2.1/entities",
4.      jsontext     = Text.FromBinary(Json.FromValue(Text.Start(Text.Trim(text), 5000))),
5.      jsonbody     = "{ documents: [ { language: ""en"", id: ""0"", text: " & jsontext & " } ] }",
6.      bytesbody    = Text.ToBinary(jsonbody),
7.      headers      = [#"Ocp-Apim-Subscription-Key" = apikey],
8.      bytesresp    = Web.Contents(endpoint, [Headers = headers, Content = bytesbody]),
9.      jsonresp     = Json.Document(bytesresp),
10. doc = jsonresp[documents]{0},
11. result = doc[entities],
12. #"Converted to Table" = Table.FromList(result, Splitter.SplitByNothing(), null, null,
        ExtraValues.Error),
13. #"Expanded Column1" = Table.ExpandRecordColumn(#"Converted to Table", "Column1",
        {"name", "matches", "wikipediaLanguage", "wikipediaId", "wikipediaUrl", "bingId",
        "type"}, {"name", "matches", "wikipediaLanguage", "wikipediaId", "wikipediaUrl",
        "type"}),
14. #"Expanded matches" = Table.ExpandListColumn(#"Expanded Column1", "matches"),
15. #"Expanded matches1" = Table.ExpandRecordColumn(#"Expanded matches", "matches",
        {"wikipediaScore", "text", "offset", "length", "entityTypeScore"}, {"wikipediaScore", "text",
        "offset", "length", "entityTypeScore"})
16. in
17.     #"Expanded matches1"
```

---

## 4. Results

During this 25-month period (i.e., from 1 January 2020 till 6 February 2022), more than 1.5 million political tweets were captured. Since our political interest lies within the Middle Eastern region, we captured tweets from regional presidents, foreign ministers, defense ministers, official ministerial accounts, political leaders, and news agencies. In total, 271,885 distinct Twitter handles were used to retrieve political tweets in 63 different languages. Of these different tweets of different languages, 578,706 tweets were in Arabic, 320,221 tweets were in English, 71,983 tweets were in French, 65,430 tweets were in Farsi/Persian, 29,103 tweets were in Spanish, and 22,219 tweets were in German, among many others. Arabic, English, French, Persian, Spanish, German, and Turkish were found to be the top seven languages with the most regional tweets. Finally, after the language detection and translation process of Microsoft Cognitive Services, sentiment analysis and NER were

performed on all of the tweets [26,27,29]. Microsoft Power BI extracted all location entities (i.e., city, continent, country region, language, and state) from the tweets of the leaders to demonstrate geospatial intelligence in ESRI ArcGIS Maps and Microsoft Bing Maps. Figures 2–10 shows geospatial intelligence extracted from the Twitter handles of selected political leaders in the ESRI ArcGIS map. Table 1 shows the details of these eight selected political leaders including Abdel Fattah Elsisi of Egypt, Barham Salih of Iraq, Fuad Hussein of Iraq, Official tweet Account of Bahrain Government, Mohammed Kareem of Palestine, Mustafa Al-Kadhimi of Iraq, Reuven Rivlin of Israel, and Faisal Bin Farhan of Saudi Arabia.

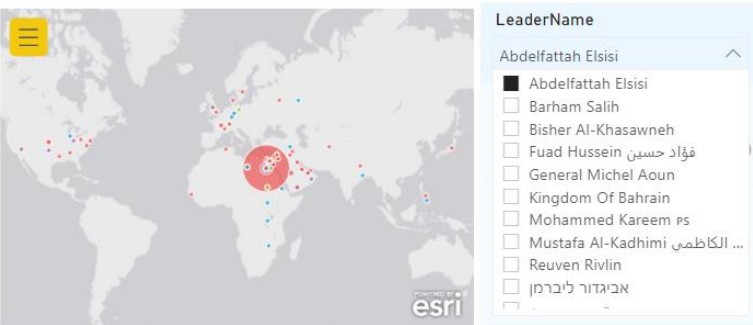

**Figure 2.** Geospatial intelligence information extracted with sentiment analysis and NER from the tweets of Abdefattah Elsisi (President of Egypt).

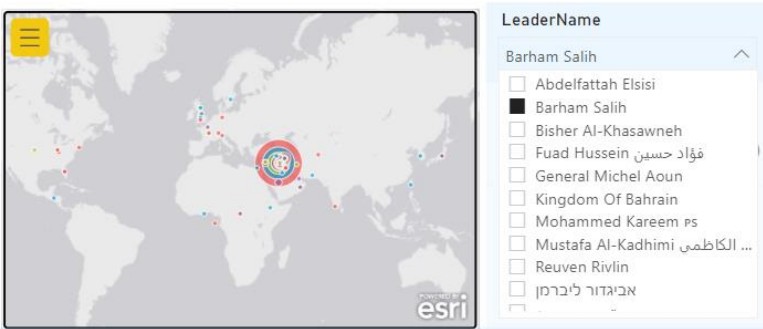

**Figure 3.** Geospatial intelligence information extracted with sentiment analysis and NER from the tweets of Barham Salih (President of Iraq).

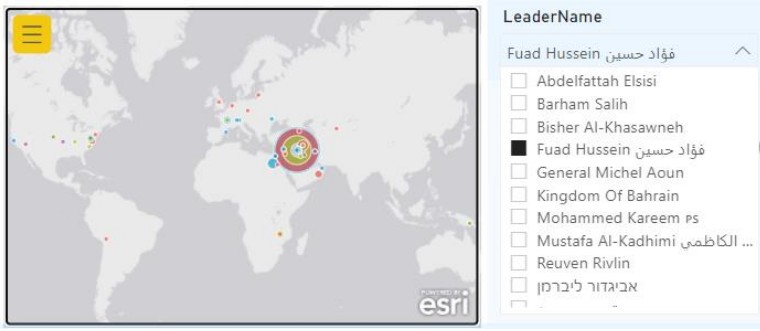

**Figure 4.** Geospatial intelligence information extracted with sentiment analysis and NER from the tweets of Fuad Hussein (Minster of Foreign Affairs, Iraq).

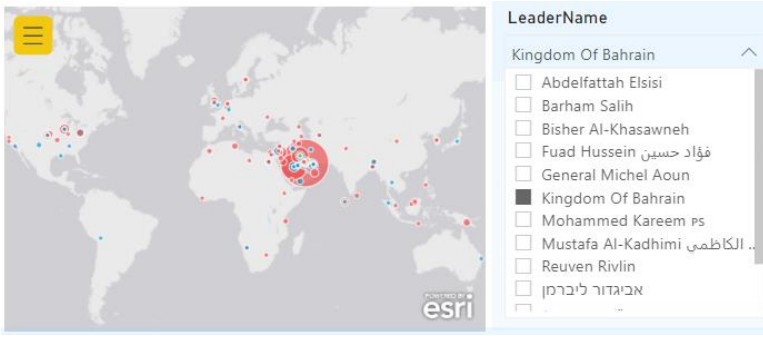

**Figure 5.** Geospatial intelligence information extracted with sentiment analysis and NER from the tweets of Fuad Hussein (Minster of Foreign Affairs, Iraq).

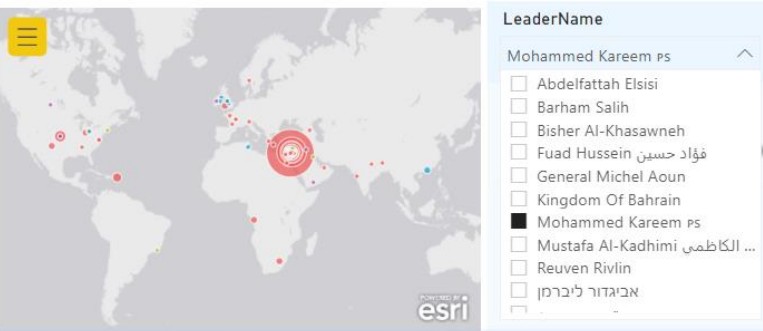

**Figure 6.** Geospatial intelligence information extracted with sentiment analysis and NER from the tweets of Mohammed Kareem (Content Creator, Palestine).

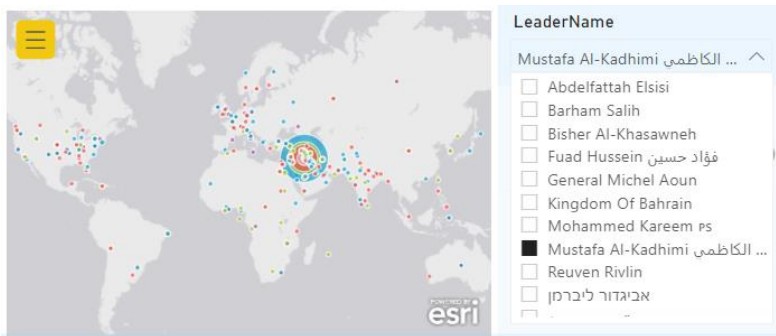

**Figure 7.** Geospatial intelligence information extracted with sentiment analysis and NER from the tweets of Mustafa Al-Kadhimi (President of Iraq).

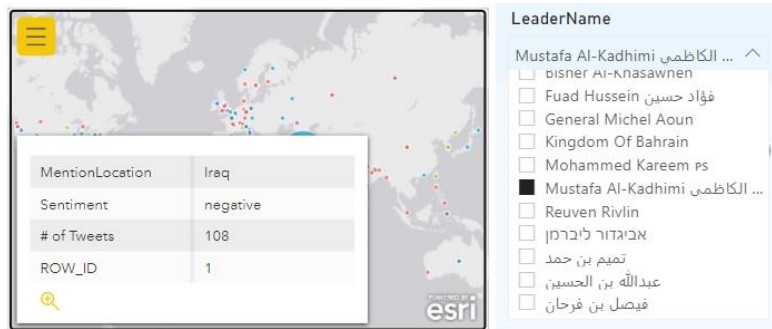

**Figure 8.** Geospatial intelligence information extracted with sentiment analysis and NER from the tweets of Mustafa Al-Kadhimi (President of Iraq).

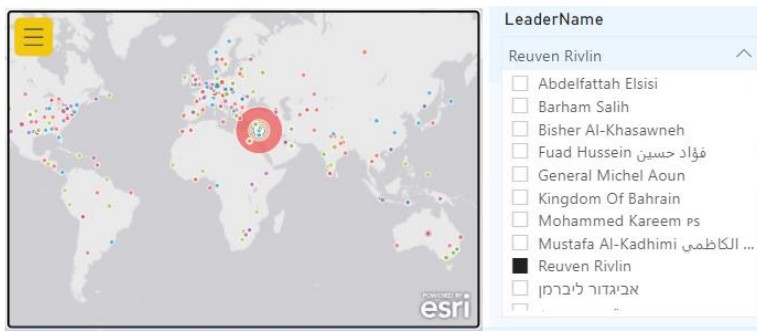

**Figure 9.** Geospatial intelligence information extracted with sentiment analysis and NER from the tweets of Reuven Rivlin (President of Israel).

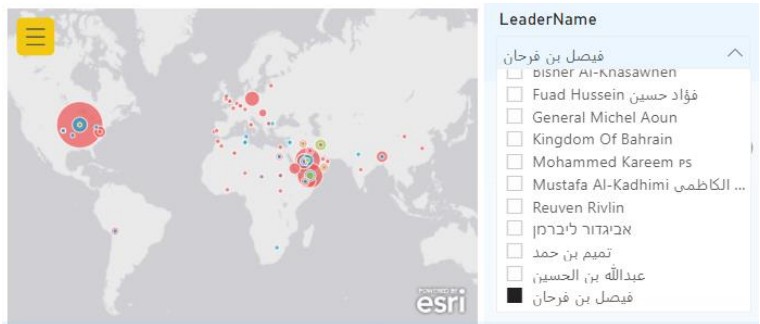

**Figure 10.** Geospatial intelligence information extracted with sentiment analysis and NER from the tweets of Faisal Bin Farhan (Minister of Foreign Affairs, Saudi Arabia).

**Table 1.** Details of 8 randomly selected political leaders out of total Twitter accounts of 271,885.

| Name | Screen Name | Country | Position | Figure Number |
|---|---|---|---|---|
| Abdelfattah Elsisi | AlsisiOfficial | Egypt | President | Figure 2 |
| Barham Salih | BarhamSalih | Iraq | President | Figure 3 |
| Fuad Hussein فؤاد حسين | Fuad_Hussein1 | Iraq | Minister of Foreign Affairs | Figure 4 |
| Kingdom of Bahrain | | Bahrain | Official tweet Account | Figure 5 |
| Mohammed Kareem PS | vic2pal | Palestine | Content Creator | Figure 6 |
| Mustafa Al-Kadhimi مصطفى الكاظمي | MAKadhimi | Iraq | President | Figures 7 and 8 |
| Reuven Rivlin | PresidentRuvi | Israel | President | Figure 9 |
| فيصل بن فرحان | FaisalbinFarhan | Saudi Arabia | Minister of Foreign Affairs | Figure 10 |

Table 2 shows the statistical details of the tweets processed for generating geospatial information, as depicted in Figures 2–10. As shown in Table 2, the tweets of Abelfattah Elisis (President of Iraq) used the most positive tones in his tweets (represented with average positive sentiment confidence of 0.62). On the other hand, Reuven Rivlin of Israel and Bahram Salih of Iraq had the highest level of negative tones represented by average negative sentiment confidences of 0.46 and 0.42, respectively. It should be mentioned that this comparison is only limited to the seven selected cases during the selected time intervals represented within Table 2 (i.e., not with the 271,885 total Tweet users profiled from 1 January 2020 to 6 February 2022 by the proposed system). Table 2 also shows the number of locations extracted (along with the unique locations) from the tweets of the leaders using the NER process of the system. It is evident from Table 2 that different political leaders represented different patterns of social media usage, with differences in frequencies of posts, sentiments of posts, and other variables.

**Table 2.** Statistics on number of tweets analyzed, sentiment analyzed, time intervals, location extracted for the selected leaders.

| Tweet Name | Number of Tweets | Average Positive Confidence | Average Negative Confidence | Average Neutral Confidence | From | Till | Total Locations Mentioned | Unique Locations Mentioned |
|---|---|---|---|---|---|---|---|---|
| Abdelfattah Elsisi | 3598 | 0.615751 | 0.191437 | 0.19281 | 6 January 2020 | 30 March 2021 | 195 | 73 |
| Barham Salih | 2409 | 0.374423 | 0.419358 | 0.233076 | 24 January 2020 | 28 March 2021 | 207 | 56 |
| Fuad Hussein فؤاد حسين | 796 | 0.247032 | 0.351935 | 0.399548 | 12 January 2020 | 14 October 2020 | 122 | 44 |
| Mohammed Kareem PS | 3736 | 0.303567 | 0.267702 | 0.22981 | 1 January 2020 | 20 March 2021 | 215 | 59 |
| Mustafa Al-Kadhimi مصطفى الكاظمي | 1538 | 0.357464 | 0.310656 | 0.339746 | 9 April 2020 | 31 March 2021 | 1460 | 307 |
| Reuven Rivlin | 2256 | 0.505517 | 0.462758 | 0.222013 | 2 January 2020 | 31 March 2021 | 1526 | 292 |
| فيصل بن فرحان | 1618 | 0.542269 | 0.266382 | 0.315602 | 6 January 2020 | 28 March 2021 | 165 | 65 |

Figure 11 shows a heat map of more than 50K locations extracted by the NER process between 12 February 2021 and 6 February 2022. This figure represents the user-mentioned location names in aggregated format for all the political leaders, campaigners, and supporters represented through 271,885 Twitter handles. Therefore, the proposed system is capable of representing the location in aggregated format (as shown in Figure 11) as well as by individual political leaders (as shown previously in Figures 2–10). It is also possible to scrutinize the location intelligence by individual political leaders within a specified time interval.

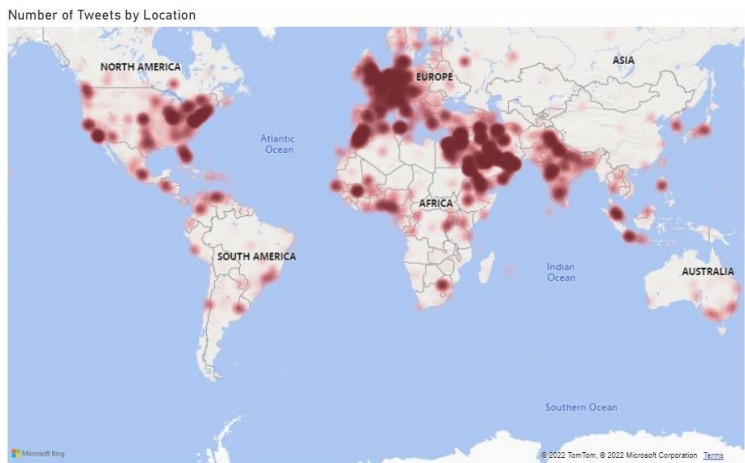

**Figure 11.** Heatmap showing more than 50K locations extracted between 12 February 2021 and 6 February 2022.

This section also demonstrated the capability of the proposed system in inspecting location intelligence by individual political leaders through different spectrums of time intervals, as shown in Figures 12–18. Figures 12–18 represent location intelligence from tweets of Binali Yıldırım. Binali Yıldırım was born on 20 December 1955 and served as the 27th Prime Minister of Turkey from 2016 to 2018. He was the leader of the Justice and Development Party (AKP) from 2016 to 2017. Our system successfully processed 498 tweets of Binali Yıldırım from his official Twitter Handle @BA_Yildirim (represented by blue tag or verified icon of Twitter) from 19 February 2021 to 6 February 2022. Out of these 498 tweets, 31 had mixed, 94 had negative, 131 had neutral, and 242 had positive sentiments. Figure 12 shows the result of processing 94 negative tweets of Binali Yıldırım, with sentiment analysis and NER. As shown in Figure 12, NER detected 17 date–time entities, 50 location entities,

93 organization entities, 32 person entities, 27 quantity entities, 1 URL entity, and 93 other entities from the 94 negative tweets of Binali Yıldırım. As shown in Figure 12, during the monitored period, Binali Yıldırım mentioned 32 persons, 50 locations, and 93 organizations through his negative tweets, as captured by our fully automated process. Figure 13 shows all of the 50 locations extracted from 94 negative tweets of Binali Yıldırım between 22 February 2021 and 5 February 2022. When the user of the system selects a specific tweet, all of the location names specified within that selected tweet are detected and displayed on a map by the proposed system, as shown in Figure 14. Finally, Figures 15–18 show the location intelligence at different time intervals.

| Type | Count of TweetText | Average of Confidence NegativeSen | Average of Confidence NeutralSen | Average of Confidence PositiveSen | RetweetCount |
|---|---|---|---|---|---|
| DateTime | 17 | 0.94 | 0.02 | 0.04 | 8049 |
| Location | 50 | 0.88 | 0.06 | 0.05 | 18300 |
| Organization | 93 | 0.92 | 0.04 | 0.03 | 34394 |
| Other | 93 | 0.92 | 0.04 | 0.04 | 47948 |
| Person | 32 | 0.92 | 0.04 | 0.04 | 7271 |
| Quantity | 27 | 0.88 | 0.09 | 0.03 | 8745 |
| URL | 1 | 0.72 | 0.19 | 0.09 | 10 |
| **Total** | **94** | **0.91** | **0.05** | **0.04** | **124717** |

**Figure 12.** Results of NER and sentiment analysis on 94 negative tweets of Binali Yıldırım (Turkish Political Leader from 22 February 2021 to 5 February 2022).

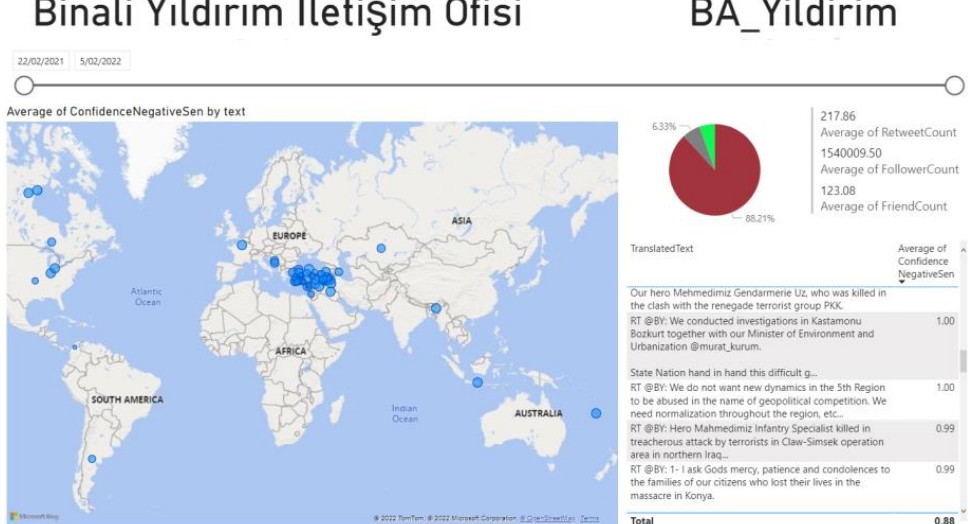

**Figure 13.** In total, 50 location names were automatically extracted from the 94 negative tweets of Binali Yıldırım from 22 February 2021 to 5 February 2022.

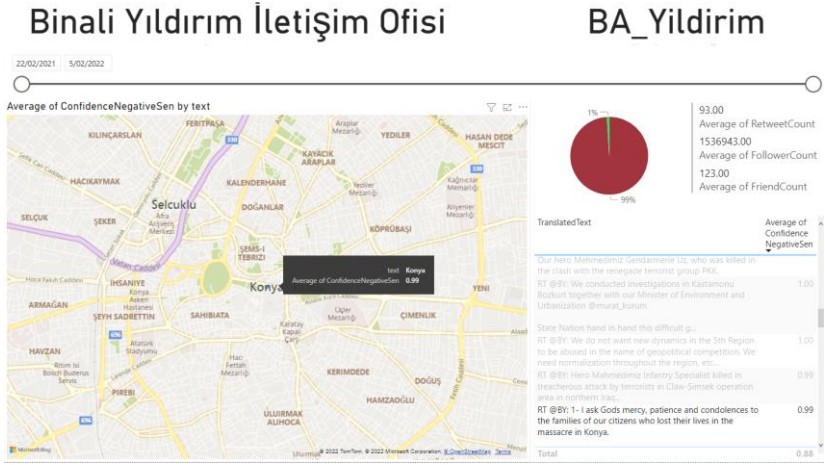

**Figure 14.** As the user of the proposed system selected a particular tweet of Binali Yildirim, the specified location (i.e., Konya) within the tweet was detected and displayed on Microsoft Bing Maps.

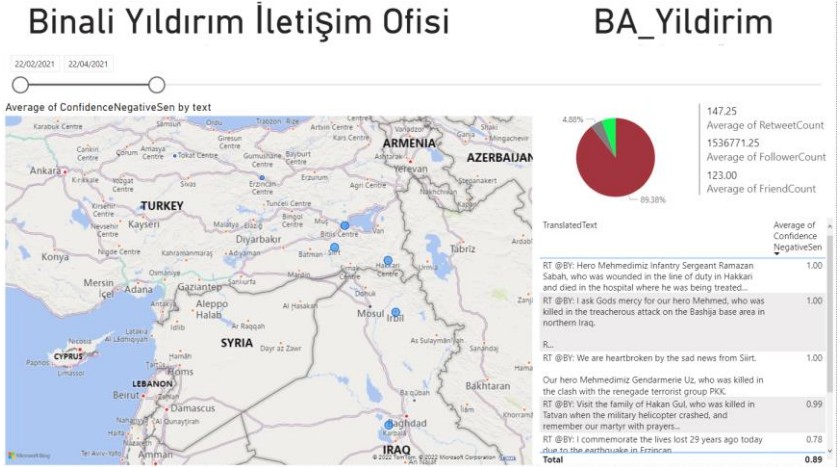

**Figure 15.** Location names automatically extracted and highlighted in Bing Maps from the negative tweets of Binali Yıldırım between 22 February 2021 and 22 April 2021.

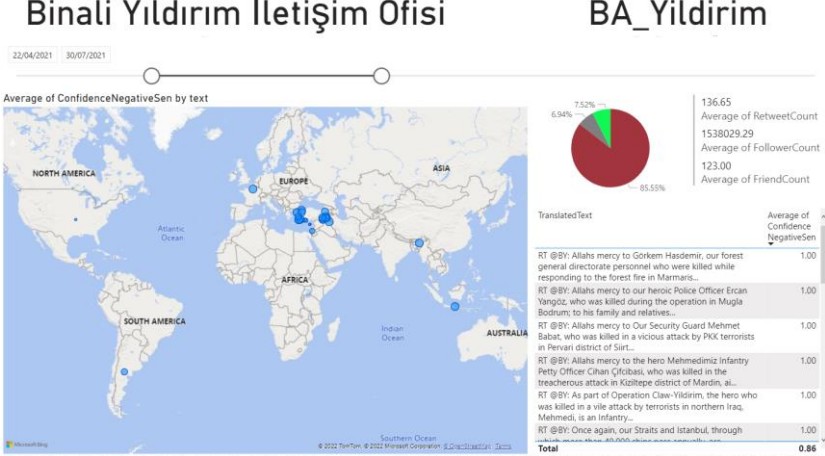

**Figure 16.** Location names automatically extracted and highlighted in Bing Maps from the negative tweets of Binali Yıldırım between 22 April 2021 and 30 July 2021.

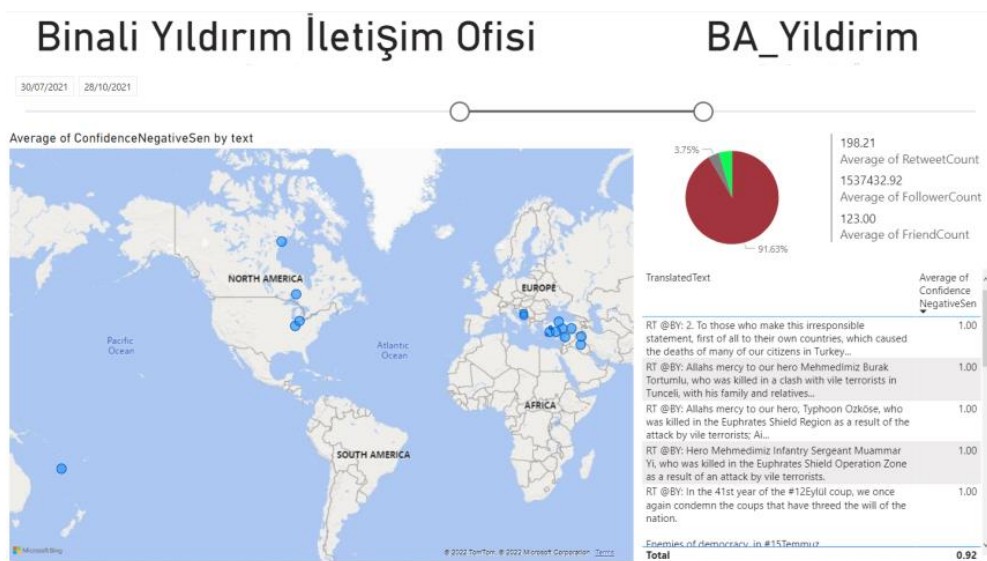

**Figure 17.** Location names automatically extracted and highlighted in Bing Maps from the negative tweets of Binali Yıldırım between 30 July 2021 and 28 October 2021.

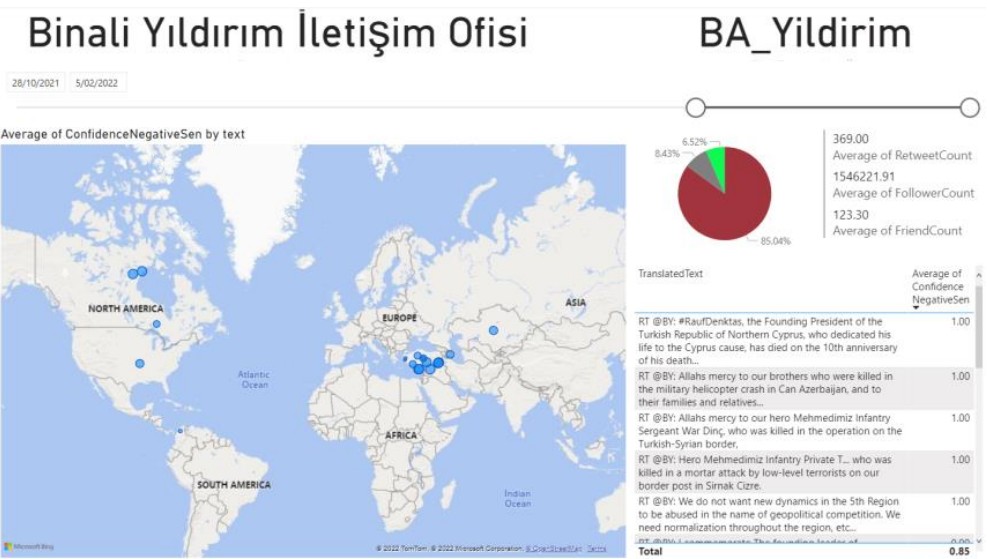

**Figure 18.** Location names automatically extracted and highlighted in Bing Maps from the negative tweets of Binali Yıldırım between 28 October 2021 and 5 February 2022.

Finally, Table 3 shows the top 10 cases of Twitter handles by the number of tweets in descending order between 12 February 2021 and 6 February 2022. As seen in Table 3, most of the tweets were captured from national news agencies in Middle Eastern countries, since these agencies reported most of the political news. It should be mentioned that Agence Tunis-Afrique-Presse demonstrated the most negative sentiment during the monitored period, as seen from Table 3.

**Table 3.** Top 10 cases of Twitter handles by the number of tweets in descending order between 12 February 2021 and 6 February 2022 along with their average sentiments.

| Twitter Handle | UserName | Number of Tweets | Confidence of Positive Sentiment | Confidence of Negative Sentiment | Confidence of Neutral Sentiment |
|---|---|---|---|---|---|
| NNALeb | National News Agency | 20,881 | 0.131523 | 0.391366 | 0.47711 |
| OmanNewsAgency | وكالة الأنباء العمانية | 17,819 | 0.123863 | 0.169898 | 0.706237 |
| Petranews | Jordan News Agency | 16,498 | 0.08586 | 0.201307 | 0.712832 |
| wamnews | وكالة أنباء الإمارات | 13,915 | 0.124672 | 0.120952 | 0.754375 |
| kuna_ar | كـــــــوناKUNA | 12,431 | 0.080433 | 0.286093 | 0.633473 |
| APS_Algerie | ALGÉRIE PRESSE SERVICE ‖ وكالة الأنباء الجزائرية | 9898 | 0.0906 | 0.17582 | 0.733578 |
| bna_ar | وكالة أنباء البحرين | 9624 | 0.189876 | 0.095896 | 0.714227 |
| tragency1 | وكالة أنباء تركيا | 6044 | 0.14363 | 0.261906 | 0.594463 |
| MAP_Information | Agence MAP | 5781 | 0.080337 | 0.163698 | 0.755964 |
| AgenceTAP | Agence Tunis-Afrique-Presse | 3021 | 0.048344 | 0.452062 | 0.499592 |

## 5. Discussion

NERs have been previously used in multiple domains, including the biomedical sector, as shown in [30]. In [30], researchers identified 14 articles that used NER exclusively within the biomedical corpus. In our previous studies, NER was used to generate locations of global events in a generic domain and not particularly focused on political science [11,12]. Moreover, in [17], our implementation of NER-based techniques was focused on the social science domain. According to the existing literature, NER has never been reported in the corpus of political science. The existing studies in political science, as seen in [4–7,21,22], have directly used location fields from Twitter feeds without applying a NER algorithm. Since this is the first study to apply a NER-based algorithm in the domain of political science, this study observed unique cases that have never been explored in previous studies [4–6,11,12,21,22,30]. During our experimentation, we witnessed several terms being incorrectly classified as location entities (i.e., false positives) and several valid location entities not being recognized as location entities (i.e., false negatives) by our implementation of NER algorithms. Moreover, there were cases where the political leader used sarcasm, and as a result, our implementation of sentiment analysis incorrectly scored the sentiment. In addition, there were a few cases in which the automated translation process mistranslated a few of the non-English tweets. As a result, subsequent processing by sentiment analysis and NER produced incorrect classifications. The effect of overall misclassification arising from these three different types of errors (i.e., incorrect classification of location, incorrect classification of sentiment due to sarcasm, and mistranslation translation) had to be measured systematically.

*Accuracy*, *precision*, *recall*, and $F_1$ *score* were measured for selected leaders that included Abdel Fattah Elsisi of Egypt, Barham Salih of Iraq, Fuad Hussein of Iraq, Mohammed Kareem of Palestine, Mustafa Al-Kadhimi of Iraq, Reuven Rivlin of Israel, and Faisal Bin Farhan of Saudi Arabia, as shown in Table 4. As shown in Table 4, the highest *accuracy* was achieved for processing the tweets of Barham Salih, with an *accuracy* of 0.99, and the lowest level of *accuracy* was scored on processing the tweets of Reuven Rivlin, with an *accuracy* of 0.94. However, the overall performances of the proposed system were noted as 0.90 *precision* rate, 0.90 *recall* rate, 0.90 $F_1$ *score*, and 0.97 *accuracy*.

**Table 4.** Detailed result of measuring performance of the proposed system with *precision*, *recall*, $F_1$ *score*, and *accuracy* against a few selected cases.

| Leader Name | TP | TN | FP | FN | Precision | Recall | F₁ Score | Accuracy |
|---|---|---|---|---|---|---|---|---|
| Abdelfattah Elsisi | 70 | 826 | 14 | 18 | 0.833333 | 0.795455 | 0.813953 | 0.965517 |
| Barham Salih | 49 | 847 | 4 | 6 | 0.924528 | 0.890909 | 0.907407 | 0.988962 |
| Fuad Hussein فؤاد حسين | 42 | 854 | 2 | 7 | 0.954545 | 0.857143 | 0.903226 | 0.990055 |
| Mohammed Kareem PS | 51 | 845 | 6 | 5 | 0.894737 | 0.910714 | 0.902655 | 0.987872 |
| Mustafa Al-Kadhimi مصطفى الكاظمي | 299 | 597 | 32 | 25 | 0.903323 | 0.92284 | 0.912977 | 0.940189 |
| Reuven Rivlin | 285 | 611 | 32 | 29 | 0.899054 | 0.907643 | 0.903328 | 0.936259 |
| فيصل بن فرحان | 59 | 837 | 8 | 9 | 0.880597 | 0.867647 | 0.874074 | 0.98138 |
| Overall | 855 | 5417 | 98 | 99 | 0.897167 | 0.896226 | 0.896696 | 0.969547 |

## 6. Conclusions

We used this innovative method of generating location information from Twitter feeds, whenever the location field or tweet is empty, or geotagged information is missing, as demonstrated in our recent research studies in [11,12,17,18]. However, none of our previous studies applied NER-based location extraction methodology in the domain of political science. In [11,12,17,18], we generated location information using NER in the domain of social science that included analysis of global events and COVID-19 situational awareness. However, this paper demonstrated a novel approach of automatically generating locations and geospatial intelligence from the Twitter feeds of political leaders, diplomats, government officials, as well as political supporters, with 97% overall *accuracy*, even if Twitter geotagged location information is missing. The proposed system utilized AI-based techniques and algorithms such as language detection and translation, sentiment analysis, and NER to dynamically represent geospatial intelligence for political scientists and researchers on a range of maps (e.g., ESRI ArcGIS Maps, Microsoft Bing Maps, etc.). Using this novel method, a political scientist or researcher can make evidence-based political decisions. This paper also demonstrated the feasibility of executing such a system for a substantially longer period (i.e., from 1 January 2020 to 6 February 2022) and the feasibility of capturing big data represented with millions of social media messages including texts, images, and videos. This study only focused on processing textual information from the tweets, and we did not process images or video files within the tweets. However, in the future, we plan to use image and video processing algorithms.

Moreover, in the future, we endeavor to use deep learning algorithms such as convolution neural network (CNN), linear regression, logistic regression, clustering algorithms such as expectation maximization (EM), similar to our previously demonstrated work in AI and machine learning [9–17].

**Author Contributions:** Conceptualization, F.S.; methodology, F.S.; software, F.S.; validation, F.S. and M.A.; formal analysis, F.S.; investigation, F.S. and M.A.; resources, F.S. and M.A.; data curation, F.S. and M.A.; writing—original draft preparation, F.S.; writing—review and editing, F.S. and M.A.; visualization, F.S.; funding acquisition, F.S. and M.A. All authors have read and agreed to the published version of the manuscript.

**Funding:** This research received no external funding.

**Institutional Review Board Statement:** Not Applicable.

**Informed Consent Statement:** Not Applicable.

**Data Availability Statement:** Not applicable.

**Acknowledgments:** The authors would like to thank Taufiqur Rahman, a development expert working for the Federal Government, Canberra, ACT, Australia, for his support during the development of dashboards for this study.

**Conflicts of Interest:** The authors declare no conflict of interest.

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
