# Peer review of "A Novel Method of Generating Geospatial Intelligence from Social Media Posts of Political Leaders"

_information, doi:10.3390/info13030120_

Round 1

Reviewer 1 Report

I feel that the paper is very well written. However, there is no reference in the discussion section. How about comparing your paper with existing articles to show your paper's originality?

Author Response

Reviewer's Comment 1: I feel that the paper is very well written.

We appreciated the generous comment of the reviewer. 

Reviewer's Comment 2: There is no reference in the discussion section. How about comparing your paper with existing articles to show your paper's originality?

We would like to thank the reviewer for this valuable suggestion. Accordingly, we have updated the manuscript and added the following new paragraph in discussion section, where we compared our study with existing literature (new references have also been added the reference section). 

"NER have been previously used in multiple domains, including biomedical as shown in [26]. In [26], researchers identified 14 articles that makes use of NER within biomedical corpus. In our previous studies, NER was used to identify location of global events in generic domain and not particularly focused in political science [10] [11]. Moreover, in [17], our implementation of NER based techniques were focused into social science domain. According to the existing literature, NER has never been reported in the corpus of political science. The existing studies in political science as seen from [4] [5] [6] [16] [19] have directly used location field from Twitter feed without applying NER algorithm. Since this is the first study to apply NER based algorithm in the domain of political science, this study observed unique cases that have never been explored in previous studies [4] [5] [6] [10] [11] [16] [19] [26]."

Reviewer 2 Report

The pseudo code of Alg 1 needs to be reformatted into the latex alg environment to look better. In general all the formattings of the code is substandard.

Up to the section of the results it is far from obvious what is novel other than the data collection effort which is not novel in itself. It appears to be a straightforward approach of the location and sentiment values. It does not look like the identification alg was novel either. What is the reported location accuracy?

Why is the accuracy component in the Discussion section? It should be in the Methodology section.

The testing of the accuracy is in relation to which proposed methodology? It is not clear if it pertains to anything novel rather than the use of it in a graphical tool.

If the purpose is to demonstrate a new tool then it needs to be stated as that. From what it seems the tool looks for the sentiment in relation to tweets across the map reporting on some aspects of the tweet features to produce sentiment etc.

What can improve the paper is to place all the methodological aspects into the methodology section. Then provide better pseudo code. Explain the 'use case' clearly and concisely with all the different aspects the user should be able to apply.

Please include these citations that discuss tweets on political behaviors which is relevant:

Chi, G., Yin, J., Smith, M. L., & Bodovski, Y. (2021). Global Tweet Mentions of COVID-19. arXiv preprint arXiv:2108.06385.

Exploring the disparity of influence between users in the discussion of Brexit on Twitter A Rajabi, AV Mantzaris, KS Atwal, I Garibay Journal of Computational Social Science 4 (2), 903-917

An LSTM model for predicting cross-platform bursts of social media activity N Hajiakhoond Bidoki, AV Mantzaris, G Sukthankar Information 10 (12), 394

Author Response

First of all, we would like to thank the anonymous reviewer for his valuable suggestions and comments. We are please to announce that using the valuable comments of the honorable reviewer as a guidance, we have thoroughly updated the manuscript. Hence, we are glad that the updated manuscript is now in a better shape than before.

Comment 1: The pseudo code of Alg 1 needs to be reformatted into the latex alg environment to look better. In general all the formattings of the code is substandard.

We appreciated the valuable suggestion of the reviewer and accordingly we have updated Algorithm 1, Code 1 and Code 2 to reflect the formatting that is expected from latex alg environment.

Comment 2: Up to the section of the results it is far from obvious what is novel other than the data collection effort which is not novel in itself. It appears to be a straightforward approach of the location and sentiment values. It does not look like the identification alg was novel either. What is the reported location accuracy?

Existing research (for example, Chi, G., Yin, J., Smith, M. L., & Bodovski, Y. (2021). Global Tweet Mentions of COVID-19. arXiv preprint arXiv:2108.06385.) merely uses the geo-tagged location information that is present within Twitter feed. If the Tweeter location field is empty, these existing studies cannot process location information. However, the presented study in this paper, generates new location information, even if the location field in twitter is empty. That is the purpose of Algorithm 1, Code 1 and Code 2. These presented algorithms generated new approximated location by analyzing Tweet Texts using a process called Named Entity Recognition (NER). None of the existing studies in the political science corpus (e.g., Exploring the disparity of influence between users in the discussion of Brexit on Twitter A Rajabi, AV Mantzaris, KS Atwal, I Garibay Journal of Computational Social Science 4 (2), 903-917) used NER to generate new location information (rather, they only used location information of the geo-tagged tweets).

For the contextual information, in our recent study we used NER in the domain of social science and the unique results are available in the following Q1 (high impact factor) journals:

  • K. Sufi and M. Alsulami, "Automated Multidimensional Analysis of Global Events With Entity Detection, Sentiment Analysis and Anomaly Detection," IEEE Access, vol. 9, pp. 152449 - 152460, 2021 (Q1, Impact Factor 3.367)
  • K. Sufi and M. Alsulami, "AI-based Automated Extraction of Location-Oriented COVID-19 Sentiments," Computers, Materials & Continua (CMC), Vols. (Accepted, in Press), pp. 1-15, 2022. (Q1, Impact Factor 3.378)

Since our presented method generated new location information, there could be classification errors in classifying new location. That is why, we conducted detailed evaluations on the location accuracy (with Recall, Precision, F1-Score, and Accuracy measures).

Comment 3: Why is the accuracy component in the Discussion section? It should be in the Methodology section.

We respected the valuable suggestion of the reviewer and accordingly we have now moved the accuracy component in the Methodology section.

However, for contextual information, we should mention that Engineering /Q1 Journals (e.g., IEEE Journals expects the accuracy component in Results and Discussion section (as opposed to Methodology section), as in the case of our following peer reviewed publications:

  • K. Sufi and M. Alsulami, "Knowledge Discovery of Global Landslides Using Automated Machine Learning Algorithms," IEEE Access, vol. 9, 2021. (Please check that accuracy component within discussion section https://ieeexplore.ieee.org/abstract/document/9546772)
  • K. Sufi and M. Alsulami, "Automated Multidimensional Analysis of Global Events With Entity Detection, Sentiment Analysis and Anomaly Detection," IEEE Access, vol. 9, pp. 152449 - 152460, 2021 (Please check that accuracy component within result section https://ieeexplore.ieee.org/abstract/document/9612169)
  • K. Sufi and M. Alsulami, "AI-based Automated Extraction of Location-Oriented COVID-19 Sentiments," Computers, Materials & Continua (CMC), Vols. (Accepted, in Press), pp. 1-15, 2022. (Accuracy component within Discussion section)

Comment 4: The testing of the accuracy is in relation to which proposed methodology? It is not clear if it pertains to anything novel rather than the use of it in a graphical tool.

We appreciate the valuable suggestion of the reviewer and accordingly we have now clarified within introduction, materials and methods, discussion, and conclusion that this research presents a new technique for generating location information from tweet text (event if the geo-tagged location information is missing). Since the proposed methodology generates approximated new location information, it needs to be evaluated for accuracy. As a result, wherever NER is used for extracting location, the location accuracy were evaluated thoroughly with Precision, Recall, F1-Score and Accuracy metrics (as in the case of https://ieeexplore.ieee.org/abstract/document/9612169).

Comment 5: If the purpose is to demonstrate a new tool then it needs to be stated as that. From what it seems the tool looks for the sentiment in relation to tweets across the map reporting on some aspects of the tweet features to produce sentiment etc.

We appreciated the valuable suggestion of the reviewer and accordingly within introduction, we have now mentioned that it is a new software tool. Moreover, we have also clarified within introduction, materials and methods, discussion, and conclusion that this research presents a new technique for generating location information from tweet text (event if the geo-tagged location information is missing). We have now clarified this with the following text within material and method section:

Since our algorithm generates new location information, we needed to assess the accuracy of newly generated location information

Comment 6: What can improve the paper is to place all the methodological aspects into the methodology section. Then provide better pseudo code. Explain the 'use case' clearly and concisely with all the different aspects the user should be able to apply.

We appreciated the valuable suggestion of the reviewer and accordingly we have thoroughly updated the manuscript.

  • Within the introduction section we have added the following new text:

Existing studies in political science domain either used the geotagged tweets or completely skipped usage of location extraction algorithms as observed from [4] [5] [6] [19] [7] [20]. For example, in [19] geotagged tweets, where location information already exist in the tweets, were only used for analysis. On the other hand, in [7], location information was never utilized, even though study in [7] was in political domain. Existing corpus of political science never utilized sophisticated location extraction mechanisms, like Named Entity Recognition (NER) that can generate location information when the geo-tagged location field is absent within Twitter feed.

Using the outcome of this research a political scientist would obtain greater level of geographic intelligence compared to the outcome offered by the previous studies in political science [4] [5] [6] [19] [7] [20].

  • Within the materials and methods section, we have added the following new text:

As mentioned before, while existing research in political science on Twitter data used geo-tagged location field of twitter data, this study research generated new locations information even if geotagged location information is missing. Using NER in a unique mechanism as demonstrated with Algorithm 1, Code 1 and Code 2, we generated location information and created heat maps demonstrating political threats and opportunities. Since our algorithm generates new location information, we needed to assess the accuracy of newly generated location information.

  • Within the material and method section we have moved the following paragraph from discussion section (as per the suggestion of the reviewer):

To assess the end-to-end performance of our algorithm, we used several evaluation metrics, including Accuracy (i.e., fraction of the total samples that were correctly classified the by the classifier), Precision (i.e., fraction of predictions as a positive class that were actually positive by the classifier), Recall (i.e., fraction of all positive samples that were correctly predicted as positive by the classifier), and the F1-score (i.e., precision and recall combined into a single measure) as represented by Equation 1 to Equation 4.       

  • Within the material and method section we have formatted Algorithm 1, Code 1 as well as Code 2 as per the valuable suggestion of the reviewer.

  • Within the Discussion section we have added the following new text:

NER have been previously used in multiple domains, including biomedical as shown in [30]. In [30], researchers identified 14 articles that used NER exclusively within biomedical corpus. In our previous studies, NER was used to generate locations of global events in generic domain and not particularly focused in political science [11] [12]. Moreover, in [17], our implementation of NER based techniques were focused into social science domain. According to the existing literature, NER has never been reported in the corpus of political science. The existing studies in political science as seen from [4] [5] [6] [7] [21] [22] have directly used location field from Twitter feed without applying NER algorithm. Since this is the first study to apply NER based algorithm in the domain of political science, this study observed unique cases that have never been explored in previous studies [4] [5] [6] [11] [12] [21] [22] [30].

  • Within the conclusion section we have added the following new text:

We have used this innovative method of generating location information from Twitter feed, whenever location field or Tweet is empty or geo-tagged information is missing as demonstrated in our recent research works in [11] [12] [17] and [18]. However, none our previous work applied NER based location extraction methodology in the domain of political science. In [11] [12] [17] and [18], we generated location information using NER in the domain of social science that included analysis of global events, and COVID-19 situational awareness.

Comment 7:

Please include these citations that discuss tweets on political behaviors which is relevant:

Chi, G., Yin, J., Smith, M. L., & Bodovski, Y. (2021). Global Tweet Mentions of COVID-19. arXiv preprint arXiv:2108.06385.

Exploring the disparity of influence between users in the discussion of Brexit on Twitter A Rajabi, AV Mantzaris, KS Atwal, I Garibay Journal of Computational Social Science 4 (2), 903-917

An LSTM model for predicting cross-platform bursts of social media activity N Hajiakhoond Bidoki, AV Mantzaris, G Sukthankar Information 10 (12), 394

We appreciated the valuable suggestion of the reviewer and accordingly we have added all the suggested references in [7], [19], and [20]. Moreover, we have briefed about these newly added references within introduction as well as the discussion section.

Round 2

Reviewer 2 Report

The edits are thorough enough